# WEB-SOBA: Word Embeddings-Based Semi-automatic Ontology Building for Aspect-Based Sentiment Classification

Fenna ten Haaf, Christopher Claassen,
Ruben Eschauzier, Joanne Tjan, Daniël Buijs,
Flavius Frasincar, and Kim Schouten

Erasmus University Rotterdam
PO Box 1738, NL-3000 DR
Rotterdam, the Netherlands
{450812fh,456177cc,480900re,413647jt,483065db}@student.eur.nl,
{frasincar, schouten}@ese.eur.nl

**Abstract.** For aspect-based sentiment analysis (ABSA), hybrid models combining ontology reasoning and machine learning approaches have achieved state-of-the-art results. In this paper, we introduce WEB-SOBA: a methodology to build a domain sentiment ontology in a semi-automatic manner from a domain-specific corpus using word embeddings. We evaluate the performance of a resulting ontology with a state-of-the-art hybrid ABSA framework, HAABSA, on the SemEval-2016 restaurant dataset. The performance is compared to a manually constructed ontology, and two other recent semi-automatically built ontologies. We show that WEB-SOBA is able to produce an ontology that achieves higher accuracy whilst requiring less than half of user time, compared to the previous approaches.

**Keywords:** Ontology learning · Word embeddings · Sentiment analysis · Aspect-based sentiment analysis

## 1 Introduction

One of the most valuable pieces of information for any business is the opinion of their customers. One source of data that can be of great help is the growing mass of online reviews posted on the Web and social media. The review forum Yelp, for example, features more than 200 million reviews about restaurants and other businesses [28]. Interpreting this massive amount of text manually would be difficult and time consuming. This is where the field of *sentiment analysis* plays an important role. It is a subfield of natural language processing (NLP), that encompasses the study of people's emotions and attitudes from written language [14]. In particular, this paper focuses on *aspect-based sentiment analysis* (ABSA), which aims to compute sentiments pertaining to specific features, the so-called 'aspects', of a product or service [22]. This results in a more in-depth sentiment analysis, as reviews can contain varying sentiment *polarities* (negative, neutral,

or positive) about different aspects. This means that through ABSA, businesses can identify which specific aspects, such as food quality or atmosphere, need to be improved upon, allowing them to make the right adjustments in their business model [19].

ABSA firstly requires that the aspects and their categories in sentences are identified (*aspect detection*) and consequently determines the sentiment with respect to these aspects (*sentiment classification*) [24]. The focus of this paper is on the sentiment classification component of sentence-level ABSA, using benchmark data where the aspects are already given [21].

Various methods have been used for sentiment classification, many of which rely on machine learning methods [8]. [24] proposes a *knowledge-based* method using domain sentiment ontologies, which represent a formal definition of the concepts that are related to a specific domain. [24] shows that using common domain sentiment knowledge encoded into an ontology gives better performance for sentiment classification, whilst requiring less training data to do so. [23] further shows that even better performance is achieved by a *Two-Step Hybrid Model* (TSHM) approach. For this approach, the first step is to use a domain sentiment ontology to predict a sentiment and, if this is inconclusive, use a machine learning algorithm that predicts the sentiment as a backup solution.

The ontologies needed as an input for these hybrid models can be obtained through different methods. One approach is to manually build an ontology [23,24], but this is time consuming and needs to be done for each domain separately. Automatic ontology construction is also an option as [3] proposes, but this process is less accurate because there is no human supervision in creating the ontology. [31] shows that a semi-automatic approach, where human input is required to control for possible mistakes made by the ontology builder, comes close to the human made ontology in accuracy whilst being more time-efficient. While the authors of [31] made use of word co-occurrences to build their ontology, the use of word embeddings has not been investigated until now for constructing a domain sentiment ontology. The advantage of word embeddings is that words are mapped to vectors, which allows for easy computation with words. Moreover, word embeddings also capture semantic features of the words such as similarity to other words. Previous authors have shown that word embeddings outperform word co-occurrence based methods for various NLP tasks [1].

In this paper, we propose a semi-automatic ontology builder called Word Embeddings-Based Semi-Automatic Ontology Builder for Aspect-Based Sentiment Analysis (WEB-SOBA). We aim to build a domain sentiment ontology from a domain corpus based on word embeddings, to exploit semantic relations between words. The source code written in Java of this project can be found at `https://github.com/RubenEschauzier/WEB-SOBA`.

The rest of the paper has the following structure. In Sect. 2 we discuss related relevant literature that forms the background of our research. In Sect. 3, an overview of the used datasets is given. Further, we describe our methodology in Sect. 4 and present our evaluation criteria and results in Sect. 5. Last, we give our conclusions and make suggestions for future work in Sect. 6.

## 2 Related Works

In this section, we provide an overview of the relevant literature on hybrid methods, ontology building, and word embeddings.

### 2.1 Hybrid Methods

The authors of [7] are among the first to suggest that a combination or 'hybrid' of knowledge-based and machine learning methods are promising in sentiment analysis. Following such a hybrid approach, [23] proposes a combination of knowledge and statistics. The used machine learning approach is the *bag-of-words* (BoW) model, where the authors train a multi-class Support Vector Machine (SVM) that is able to classify an aspect into one of three sentiment values: negative, neutral, or positive. The authors show that using a BoW model only as a backup when making predictions using an ontology (Ont+BoW) results in an improvement compared to alternative models.

Similar to the two-stage approach in [23], [27] uses a combination of methods in a framework called HAABSA, to predict the sentiment values in sentence-level ABSA. Instead of using the BoW model, the authors use a Left-Center-Right separated neural network with Rotatory attention (*LCR-Rot*) model from [30]. [27] finds that an alteration of the LCR-Rot model (*LCR-Rot-Hop*) as the backup model, where the rotatory attention mechanism is applied multiple times, has the highest performance measure and is even able to outperform the Ont+BoW model of [23]. For this reason we favor using this approach to evaluate the performance of our ontology.

### 2.2 Ontology Building

As described by [6], there are various subtasks associated with the development of an ontology. The first step is to gather linguistic knowledge in order to be able to recognize domain-specific terms as well as synonyms of those terms. All terms with the same meaning need to be clustered together to form concepts (e.g., 'drinks' and 'beverage' can both be a *lexicalization* of the concept Drinks). In addition, hierarchical relationships need to be established (e.g., given the class Food and the class Fries, Fries should be recognized as a subclass of Food). Next, non-hierarchical relations between concepts are defined, as well as certain rules in order to be able to derive facts that are not explicitly encoded by the ontology.

A manually built ontology is given in [23]. Since this ontology was made manually, it has great performance by design. However, building the ontology requires a lot of time. [31] shows that using a semi-automatically built ontology substantially decreases the human time needed to create an ontology, while having comparable results to benchmark models. The authors of [31] focus on using word frequencies in domain corpora for ontology building. [9] further extends this work by making use of *synsets*, or sets of synonyms, in term extraction, concept formation, and concept subsumption. However, differently than [9] and [31], we focus on using *word embeddings* for the automated part of the ontology

building, meaning that words are mapped to vectors that retain certain similarities between words. As discussed in the previous section, [1] shows that word embeddings outperform word co-occurrences for certain NLP tasks. We hypothesize that word embeddings can be effective for ontology building based on this previous work.

## 2.3   Word Embeddings

A word embedding is a method for mapping various words to a single vector space. It creates vectors in a way that retains information about the word the vector represents, whilst having relatively low dimensionality when compared to the bag-of-words approach.

Some of the most well-known methods for word embedding are proposed by [16], known as *local context window methods*. These methods primarily consider a word within the local context it appears, such that the vector of the word is determined by its sentence-level neighbours. The authors introduce the Continuous Bag-of-Words model (CBOW) and the Skip-gram model. It is shown that these local window context methods outperform previous *global matrix factorization* methods like Latent Semantic Analysis (LSA) and Latent Dirichlet Allocation (LDA) [16]. CBOW and Skip-gram are not only able to represent words, but can also detect syntactic and semantic word similarities. Relations like Athens→Greece are established by training on large text files that contain similar relations, e.g., Rome→Italy. Implementations of CBOW and Skip-gram are publicly available in the 'word2vec' project.

A different method for embedding words was introduced in response to CBOW and Skip-gram. [20] combines global factorization and local context window methods in a bilinear regression model called the Global Vector method, abbreviated as GloVe. GloVe produces word embeddings by primarily considering non-zero word co-occurrences of the entire document.

A last method for word embedding, called FastText, is introduced by [4]. FastText extends the Skip-gram model by including 'subword' information. The advantage of such an approach is that a vector representation of an unknown word can be formed by concatenating words, e.g., the vector for 'lighthouse' is associated with the vector for 'light' + the vector for 'house'.

There is no well-defined 'best' word embedding amongst word2vec, GloVe, and FastText for all NLP tasks. Some authors suggest that the difference in performance is mainly due to differences in hyperparameter settings between methods [13]. Other authors suggest that the word embedding methods are similar in practice, as can be found in the results of [18] and [25], for example. There are some exclusive features for each method, however. As an example, FastText can generate a word embedding for a word that does not exist in the database. On the other hand, word2vec is training time efficient and has a small memory footprint. For these practical considerations, we opt for the word2vec algorithm in our research.

## 3  Data

In sentiment analysis, there are a number of standard datasets that are widely used. We focus on datasets for the restaurant domain, because this is also the domain based upon which the ontologies from [9], [23], and [31] were built, lending for easier comparison.

To create an ontology, we need a domain-specific corpus and a contrasting corpus, in order to find how frequent certain words appear in a domain, relative to general documents. The domain-specific corpus is created using the Yelp Open Dataset [28]. This dataset consist of consumer reviews of various types of businesses. We filter out the reviews that are not about restaurants, resulting in with 5,508,394 domain-specific reviews of more than 500,000 restaurants. For the contrasting corpus, we use the pre-trained word2vec model google-news-300 [12], containing vectors for 3 million words with a dimensionality of 300.

To evaluate our ontology, we use the SemEval-2016 Task 5 restaurant data for sentence-level analysis [21]. It is a standard dataset that contains restaurant reviews. It is structured per review and each review is structured per sentence, with reviews having varying amounts of sentences. There are 676 sentences in the test set and 2000 sentences in the training set. Each sentence contains opinions relating to specific `targets` in the sentence. The `category` of a target is also annotated, which is made up of an entity $E$ (e.g., restaurant, drinks) and attribute $A$ (e.g., prices, quality) pairing E#A. Furthermore, each identified E#A pair is annotated with a polarity from the set $\mathcal{P} = \{$negative, neutral, positive$\}$.

When an entity is only implicitly present in a sentence, the target is labeled as `NULL`. Since most machine learning methods need a target phrase to be present, the implicit sentences are not used in the analysis. These implicit aspects make up around 25% of the training set, leaving still 1879 explicit aspects. Figure 1 gives an overview of the aspects and polarities labeled in the dataset.

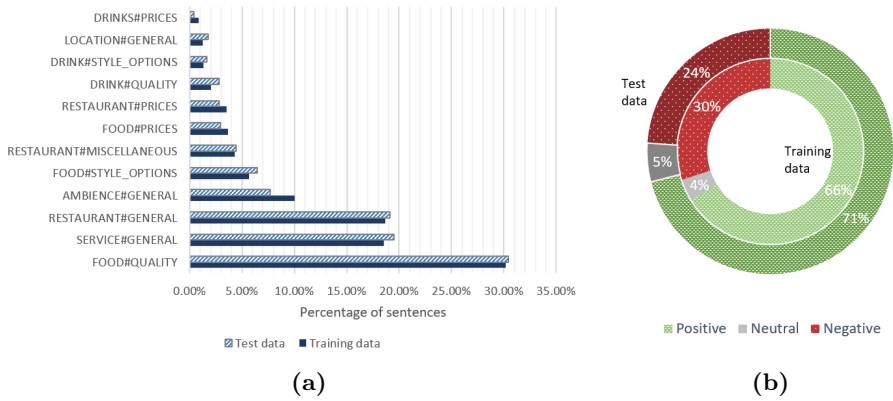

**(a)**  **(b)**

**Fig. 1.** Percentage of occurrence for aspects per sentence (a) and polarities per aspects (b) for the SemEval-2016 Task 5 restaurant dataset.

# 4 Methodology

In this section, we explain how our ontology builder works, which we refer to as the Word Embedding-Based Semi-Automatic Ontology Builder for Aspect-Based Sentiment Analysis (WEB-SOBA). Additionally, we discuss the user input required at various points in the ontology building process.

## 4.1 Word Embeddings

The word embedding method we use for our ontology building is word2vec, which uses a two-layer neural network [17]. There are two variations to this. In the Continuous Bag-of-Words (CBOW) model, the embedding is learned by predicting the current word based on its context (the surrounding words). Another approach is to learn by predicting the surrounding words given a current word. This approach is called the Skip-gram model. CBOW is better for frequent words, while Skip-gram does a better job for infrequent words. Moreover, CBOW can be trained faster than Skip-gram. However, in practical applications the performance is fairly similar. For this paper, due to the previously given advantages, we use the CBOW model to make the word embeddings. This model is trained using the following loss function:

$$J = \frac{1}{T} \sum_{t=1}^{T} \log p(w_t | w_{t-c}, \ldots, w_{t-1}, w_{t+1}, \ldots, w_{t+c}), \tag{4.1}$$

where $[-c, c]$ is the word context of the word $w_t$ and T represents the number of words in the sequence.

## 4.2 Ontology Framework

The first step to build our ontology is to decide upon the basic structure that will be used. We use the same structure as the ontology presented by [23]. This ontology contains two main classes: `Mention` and `SentimentValue`. `SentimentValue` consists of the subclasses `Positive` and `Negative`. Please note that our ontology does not model neutral sentiment. This is a deliberate choice, as neutral sentiment has an inherent subjectivity [23]. The skeletal structure already starts with a certain number of `Mention` subclasses based on the entities and attributes that make up categories within the domain, denoted as `ENTITY#ATTRIBUTE` (e.g., `FOOD#QUALITY`). For the considered restaurant domain, the `Mention` subclasses are: *Restaurant, Location, Food, Drinks, Price, Experience, Service, Ambiance, Quality*, and *Style&Options*. In addition we also use the attributes *General* and *Miscellaneous*, but we do not explicitly represent them in sub-classes. Figure 2 illustrates the `Mention` subclasses in our base ontology and how they are related to each other to make `ENTITY#ATTRIBUTE` pairs.

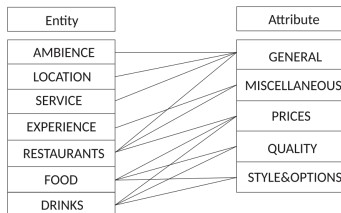

**Fig. 2.** Entities and Attributes, as E#A pairs.

In this ontology we make a distinction between the part-of-speech of a term, i.e., whether it is a verb (`Action`), noun (`Entity`), or adjective (`Property`). The `Mention` classes are superclasses of classes denoting the aspect and the part-of-speech. For example, `PricesPropertyMention` is a subclass of `PricesMention` and `PropertyMention`. Each of these classes then contains classes which are a subclass of the corresponding `Mention` class and the `SentimentValue` class. For example, a class `LocationPositiveProperty` is a subclass of the classes `LocationPropertyMention` and `Positive`.

There are three types of sentiment-carrying classes: Type-1 (generic sentiment), Type-2 (aspect-specific sentiment), and Type-3 (context-dependent sentiment). The first type always has the same sentiment value regardless in what context the concept is found (e.g., the concept 'good' is always positive). This type of Mention subclasses are also immediately a subclass of the `GenericPositive` or `GenericNegative` classes, which are subclasses of the `SentimentValue` classes `Positive` and `Negative`, respectively. Next, sentiment Type-2 words only apply to a specific category of aspects and with the same sentiment. For example, the word 'delicious' applies to food and drinks but is never used when talking about service. Therefore, when determining the sentiment of an aspect in the service category, this word could be ignored. Finally, Type-3 words indicate different sentiments depending on the aspect category. For example, 'cold' beer is positive whereas 'cold' pizza is negative.

All concepts in the ontology can have two types of properties. The `Mention` classes can first of all have one or multiple aspect properties through which they are linked to all the aspect categories for which they are applicable. For example, `FoodMention` is linked to `FOOD#PRICES`, `FOOD#QUALITY`, and `FOOD#OPTIONS`. Second, differently than the main `Mention` and `SentimentValue` classes, the classes have lexical representations attached.

**User Intervention.** The user helps initialise the skeletal ontology by providing some lexicalizations for the base classes. In particular, we add some Type-1 sentiment words, such as 'enjoy', 'hate', 'great', and 'bad'. For each of these terms, the 15 most similar words obtained through word embeddings are suggested to the user. The user can then accept or reject adding these words as an extra lexicalization of the corresponding class. The reason to initialise the ontology in this

manner is that these generic, context-independent Type-1 words such as 'enjoy' and 'hate' do indicate a sentiment, but are used in a wide range of contexts. Therefore, these words are less likely to be extracted when determining terms specific to the restaurant domain, yet they are still useful if they are added to our ontology.

### 4.3 Term Selection

To first gather domain-specific words, we use the Yelp dataset [28] containing text about our domain and extract all adjectives, nouns, and, verbs from the text using the *part-of-speech* tagger of the Stanford NLP Processing Group [26]. Adverbs can also be considered for sentiment information, but they sometimes affect the intensity of the sentiment rather than the polarity [11]. Consequently, we only use adjectives, nouns, and verbs for our analysis, as they carry the primary sentiment information of the sentence. Afterwards, we select certain terms from the list of domain-specific words that are to be proposed to the user. We first assign a *TermScore* (TS) to each word, calculated using the *DomainSimilarity* (DS) and the *MentionClassSimilarity* (MCS).

The DS can be computed using the *cosine similarity*. The function for DS of word $i$ is:

$$DS_i = \frac{v_{i,D} \cdot v_{i,G}}{\|v_{i,D}\| \cdot \|v_{i,G}\|}, \tag{4.2}$$

where $v_{i,D}$ is the vector of word $i$ in the domain-related word embedding model and vector $v_{i,G}$ is generated for word $i$ in the general word embedding model. If a word is domain-specific, the cosine similarity between $v_{i,D}$ and $v_{i,G}$ is low which indicates a low value for DS as well.

The MCS calculates the maximum similarity for word $i$ to one of the *Mention* classes. The function for MCS of word $i$ is:

$$MCS_i = \max_{a \in \mathcal{A}} \left( \frac{v_i \cdot v_a}{\|v_i\| \cdot \|v_a\|} \right), \tag{4.3}$$

where $\mathcal{A}$ is the set of *Mention* classes, which are *Restaurant, Location, Food, Drinks, Price, Experience, Service, Ambiance, Quality,* and *Style&Options*. The more similar terms are to the lexical representation of the name of one of the mention classes, the higher the value of MCS.

The function TS is defined as the harmonic mean of DS and MCS. Because we want the $DS_i$ value as low as possible, we take its reciprocal value. Both DS and MCS play an important role, because we want words to be both specific to the domain (as represented by DS), and at the same time we want them to be important to the base classes in our domain (as measured by MCS). The function for the TS of each term $i$ is:

$$TS_i = \frac{2}{DS_i + MSC_i^{-1}}, \tag{4.4}$$

A specific threshold parameter is used, one for each of the three lexical classes (adjectives, nouns, and verbs), to determine whether to suggest terms to the user.

A term will be selected and proposed to the user if its TS exceeds the threshold of the lexical class, which the term belongs to. A critical issue here is deciding the value of the threshold parameter. The value of this parameter determines the amount of terms the user is asked to review. By setting the value of the threshold too low, a lot of terms have to be considered by the user, which is very time consuming. When the value of the threshold parameter is too high, crucial words will be omitted and thus absent from the ontology. The threshold value is defined for lexical class $lc$ (part-of-speech) as follows:

$$TH_{lc}^* = \max_{TH_{lc}} \left( \frac{2}{\frac{n}{accepted} + \frac{1}{accepted}} \right), \tag{4.5}$$

where $TH_{lc}$ is a threshold score for lexical class $lc$, $n$ is the number of suggested terms, and *accepted* is the number of accepted terms. $TH^*$ is defined per lexical class so that we maximize the number of accepted terms and the number of accepted terms relative to suggested terms for each lexical class.

**User Intervention.** After the TermScore of a term exceeds the threshold and the term is suggested to the user, the user can accept or reject whether to add this term to the list of all relevant terms for the ontology. If the term is a verb or a noun, the user decides if the term refers to an aspect, or if it refers to a sentiment. Adjectives are always treated as denoting a sentiment. If the term is added as a *Sentiment Mention*, the user has to decide if the word is a Type-1 *Sentiment Mention* or not, and if so, if the word is positive or negative.

Furthermore, we select all words for each accepted term that are similar to the accepted term using word embeddings. If the cosine similarity is larger than a certain threshold, the word is added to the ontology. We find in preliminary research that a threshold of 0.7 ensures that the vast majority of words added are valuable to the ontology.

## 4.4 Sentiment Term Clustering

After selecting the important terms for our ontology and letting the user define whether the terms are *Sentiment Mentions* or *Aspect Mentions*, we can create a hierarchy for the words that were deemed to be sentiment words. In this case, we want to determine the `SentimentValue` of each word, as well as determine which `Mention` class(es) the word belongs to if it is not a `GenericPositive` or `GenericNegative` sentiment.

A drawback of using word2vec word embeddings for our application is that the generated vectors do not directly account for sentiment. For example, the vectors for 'good' and 'bad' are very similar to each other because they appear in the same context, even though they convey a different sentiment. This complicates the process of determining the sentiment for our *Sentiment Mention* words. Our proposed solution to this problem is to refine our existing word2vec model, as trained on the Yelp dataset, by making the vectors sentiment-aware.

[29] uses the Extended version of Affective Norms of English Words (E-ANEW) as a sentiment lexicon. This sentiment lexicon is a dataset that attaches emotional ratings to words. Using this sentiment lexicon, the authors find and rank the $k$ (e.g., $k = 10$) most similar words in terms of emotional ratings to the target word that needs to be refined, where the most similar word gets the highest rank. These words are called the neighbours of our target word. The neighbours of our target word are then also ranked in terms of their cosine similarity, where again the word with the highest similarity gets the highest rank. After creating these sentiment and similarity rankings, the vector of the target word is refined so that it is: (1) closer to its neighbours that are sentimentally similar, (2) further away from dissimilar neighbors, and (3) still relatively close to the original vector.

Using these vectors we cluster our *Sentiment Mention* terms. For each term we calculate the cosine similarity between all of our base `Mention` classes and rank them in descending order. Additionally, we calculate the negative and positive score of our *Sentiment Mention* term in the following way:

$$PS_i = \max_{p \in P} \left( \frac{v_i \cdot v_p}{\|v_i\| \cdot \|v_p\|} \right) \qquad NS_i = \max_{n \in N} \left( \frac{v_i \cdot v_n}{\|v_i\| \cdot \|v_n\|} \right) \qquad (4.6)$$

where $PS_i$ and $NS_i$ are the positive and negative score for $term_i$. $\mathcal{P}$ and $\mathcal{N}$ are a collection of positive and negative words that span different intensities of positivity and negativity, respectively. The set of negative words is as follows: $\mathcal{P} = \{$'good', 'decent', 'great', 'tasty', 'fantastic', 'solid', 'yummy', 'terrific'$\}$. The set of positive words is: $\mathcal{N} = \{$'bad', 'awful', 'horrible', 'terrible', 'poor', 'lousy', 'shitty', 'horrid'$\}$. Finally, we have $v_i$, $v_p$, and $v_n$ which are the word embeddings of word $i$, $p$, and $n$, respectively. We predict our $term_i$ to be positive if the $PS_i$ is higher than $NS_i$. If the $PS_i$ is lower than $NS_i$, the $term_i$ is considered to be negative.

**User Intervention.** The user is asked for each *Sentiment Mention* term if it can refer to the base `Mention` class that it has the highest cosine similarity to. If the user accepts the term into the recommended `Mention` class, the user is asked then to confirm if the predicted polarity of the *Sentiment Mention* is correct. Thereafter, the user is asked the same for the `Mention` class that has the second highest cosine similarity to our term. This continues until either all `Mention` classes are accepted or one `Mention` class is rejected. After the process terminates, the *Sentiment Mention* term is added to the ontology in accordance to the decisions made by the user. Each *Sentiment Mention* can be added to multiple `Mention` classes, because a *Sentiment Mention* can convey sentiment for multiple `Mention` classes. An example of this is the word 'idyllic', which can convey sentiment about the restaurant, location, experience, and ambiance.

### 4.5    Aspect Term Hierarchical Clustering

The next step is clustering and building the ontology's hierarchy for the *Aspect Mention* terms. As previously stated, words can be represented in a vector space using word embeddings, which means it is possible to cluster these terms.

Building the ontology's hierarchy is done in two steps. First, we implement an adjusted k-means clustering approach to cluster the accepted terms into clusters corresponding to the base `Mention` classes. These clusters are *Restaurant, Location, Food, Drinks, Price, Experience, Service, Ambiance, Quality,* and *Style&Options.* Since the base mention classes are known, we can just add each data point to the base cluster with which it has the highest cosine similarity (using the lexical representations corresponding to the names of these clusters). After clustering the terms into the base subclasses, the next step is building a hierarchy for each subclass using agglomerative hierarchical clustering. Terms start in a single cluster and are slowly merged together per iteration based on a linkage criteria. The method we choose for implementation is called Average Linkage Clustering, abbreviated as ALC, as it is less sensitive to outliers. It is defined as:

$$ALC(A, B) = \frac{1}{|A| \cdot |B|} \sum_{a \in A} \sum_{b \in B} d(a, b) \tag{4.7}$$

where $d(a, b)$ is the Euclidean distance between vectors $a$ and $b$, where $a$ is in cluster $A$ and $b$ is in cluster $B$. At each iteration, terms with the lowest ALC value are clustered, creating the required hierarchy. For our ALC algorithm, we make use of the implementation described by [2]. Based on preliminary experiments of implementing the elbow method, the maximum depth that our dendrogram can possibly have is set to three for each subclass.

**User Intervention.** For each `Mention` class, each term belonging to that cluster is presented to the user and the user can accept or reject it. If the user rejects it, the user is prompted to specify the right cluster. By doing this, all terms start in the correct cluster before building a hierarchy.

## 5    Evaluation

In this section, we discuss the procedure by which we evaluate our proposed ontology. We compare the performance of our ontology with benchmark (semi-) manually built ontologies. In Sect. 5.1 we describe the performance measures we use for evaluation. Next, in Sect. 5.2 and Sect. 5.3 we present our results. All results were obtained on an Intel(R) Core(TM) i5-4690k CPU in combination with an NVIDIA GeForce GTX 970 GPU and 16 GB RAM.

### 5.1    Evaluation Procedure

The output of our proposed methodology is a domain-specific sentiment ontology based on word embeddings. This ontology is used in combination with machine

learning methods to classify the sentiment of aspects. We evaluate the quality of our proposed method by looking at the time required to construct the ontology and its performance for sentiment classification. To determine which configuration performs better, we use the Welch t-test to compare the cross-validation outcomes.

As we build the ontology semi-automatically, it is interesting to consider the time required to make a WEB-SOBA ontology. The ontology building time can be divided in human time spent and computer time spent. We try to minimize the total time spent on building the ontology, but value time spent by humans as more expensive than time spent by computers. Ultimately, we expect that our ontology performs better than the ontology of [9] and [31] if the word embeddings truly outperform word co-occurrences. However, we expect the best performance from the ontology of [23] as this ontology was entirely made by hand to perform best.

The experiments are executed on the HAABSA implementation [27]. The experimental setup for testing our ontology is simple, as we can directly plug our ontology into the HAABSA code after it has been semi-automatically constructed. We test the following ontologies: Manual [23], SOBA [31], SASOBUS [9], and WEB-SOBA. These ontologies are evaluated when used by themselves and when used in conjunction with the LRC-Rot-Hop backup model in the HAABSA framework.

## 5.2 Ontology Building Results

We now evaluate the WEB-SOBA ontology. In the end, 376 classes are added to our ontology, of which there are 15 Type-1 sentiment words, 119 Type-2 sentiment words, and 0 Type-3 sentiment words. These results are not fully unexpected: there are few Type-1 sentiment words, because these words are often not domain-specific and therefore less likely to be selected by the term selection algorithm. There are also no Type-3 sentiment words selected, possibly because there are not many of these in the dataset. To put it into perspective, the manual ontology of [23] has only 15 Type-3 sentiment words. Table 1 presents the distribution of classes, lexicalizations, and synonyms in our ontology, compared to the other benchmark ontologies.

**Table 1.** Distribution of ontology classes and properties for the manual ontology, SOBA, and WEB-SOBA.

|                 | Manual | SASOBUS | SOBA | WEB-SOBA |
|-----------------|--------|---------|------|----------|
| Classes         | 365    | 558     | 470  | 376      |
| Lexicalizations | 374    | 1312    | 1087 | 348      |

It is clear from Table 1 that our ontology is not as extensive compared to the other three ontologies. However, this does not necessarily mean that the ontology

should have a worse performance, as it is possible that the most important terms of the domain are captured. Another important factor to consider is the building time duration. Table 2 presents the time that is required for user input and for computing in minutes, compared to benchmark ontologies.

**Table 2.** Time duration of the building process for the manual ontology, SOBA, and WEB-SOBA.

|  | Manual | SASOBUS | SOBA | WEB-SOBA |
|---|---|---|---|---|
| User time (minutes) | 420 | 180 | 90 | 40 |
| Computing time (minutes) | 0 | 300 | 90 | 30 (+300) |

Table 2 shows that with respect to user time required, WEB-SOBA clearly outperforms the other semi-automatic ontology builders, and, obviously, the manual ontology as well. The computing time for running the code between user inputs is only 30 minutes, which is also lower than the computing time required for the SOBA and SASOBUS ontologies. However, the word embeddings need to be created the first time the program is used and the terms in the Yelp dataset have to be extracted, which requires an additional 300 minutes of computing time. After the vectors have been made, they can be reused in the same domain. This substantially decreases the time needed to do a different NLP task in the same domain. Additionally, the computation times are higher because the dataset used for this ontology contains millions of reviews compared to the 5001 reviews used to create SOBA and SASOBUS.

### 5.3 Evaluation Results

We compare the results of WEB-SOBA with the results of the Manual ontology from [23], the SOBA ontology of [31], and the SASOBUS ontology of [9]. Table 3 presents the performance of the ontologies by themselves, while Table 4 presents the results of the Ontology+LCR-Rot-Hop approach [27]. The tables additionally present p-values for the Welch t-test to test for equal means (under unequal variances) for the cross-validation accuracies.

**Table 3.** Comparison results for different ontologies by themselves on SemEval-2016 Task 5 restaurant data.

| | Out-of-Sample | In-Sample | Cross-validation | | Welch t-test | | | |
|---|---|---|---|---|---|---|---|---|
| | Accuracy | Accuracy | Accuracy | St. dev. | Manual | SASOBUS | SOBA | WEB-SOBA |
| Manual | **78.31**% | **75.31**% | **74.10**% | 0.044 | - | | | |
| SASOBUS | 76.62% | 73.82% | 70.69% | 0.049 | 0.118 | - | | |
| SOBA | 77.23% | 74.56% | 71.71% | 0.061 | 0.327 | 0.685 | - | |
| WEB-SOBA | 77.08% | 72.11% | 70.50% | 0.050 | 0.107 | 0.636 | 0.935 | - |

**Table 4.** Results for ontologies on SemEval-2016 Task 5 restaurant data, evaluated with LCR-Rot-Hop model as backup.

| | Out-of-Sample | In-Sample | Cross-validation | | Welch t-test | | | |
|---|---|---|---|---|---|---|---|---|
| | Accuracy | Accuracy | Accuracy | St. dev. | Manual | SASOBUS | SOBA | WEB-SOBA |
| Manual | 86.65% | 87.96% | 82.76% | 0.022 | - | | | |
| SASOBUS | 84.76% | 83.38% | 80.20% | 0.031 | 0.052 | - | | |
| SOBA | 86.23% | 85.93% | 80.15% | 0.039 | 0.088 | 0.975 | - | |
| WEB-SOBA | **87.16%** | **88.87%** | **84.72%** | 0.017 | 0.043 | 0.001 | 0.005 | - |

The tables presented above show that WEB-SOBA achieves similar performance to other semi-automatically built ontologies when evaluated by itself, even though it required less than half of the user time to build. When combined with the LCR-Rot-Hop backup method, the WEB-SOBA ontology performs better than all other ontologies including the manual one (at 5% significance level). This shows that our ontology complements the machine learning method well.

Our approach is generalizable to other domains as well. For another domain one only needs to provide the following items: a text corpus, the category classes, and the positive/negative word collections. Even if the overall time is comparable with the one of a manual approach, one major benefit for our approach is that most of the time is spent by system computations instead of user work. As limitations, our solution is not able to deal with polysemous words and our data set was not able to provide for Type-3 sentiment words.

## 6 Conclusion

For this research paper, we used word embeddings to semi-automatically build an ontology for the restaurant domain to be used in aspect-based sentiment analysis. The ontology builder we propose reduces both computing and user time required to construct the ontology, given that the word embeddings for a specific domain are already made. Furthermore, our method requires less user time compared to recently proposed semi-automatic ontology builders based on word co-occurrences, whilst it achieves similar or better performance. Our proposed ontology even reaches higher accuracy compared to the performance of a manually built one when implemented in a hybrid method with the LCR-Rot-Hop model as backup.

As future work we would like to analyse for which aspects the proposed approach gives the best performance. Also, we suggest that the performance of an ontology used for ABSA can likely be improved by using a contextual word embeddings method, such as BERT [10]. These contextual methods assign a different vector to the same word in a different context. We expect that these contextual word embeddings lead to more accurate sentiment classification, as the various meanings of words are taken into account. However, these methods cannot be directly implemented into the current ontology structure, as the same word gets a different vector in a different linguistic context. As future work

we propose to extend the ontology structure by allowing the same word to appear for multiple concepts by conditioning its presence on a certain meaning. In addition, we plan to use domain modelling [5] that complements well our contrastive corpus-based solution by providing domain-specific terms that are more generic than the current ones. Furthermore, the clustering of terms can be improved. Our proposed clustering method works for a relatively moderate amount of domain terms, but is not feasible for a huge number of domain terms. For an efficient hierarchical clustering method for Big Data, one could exploit the BIRCH algorithm as in [15].

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
