# OpenReview forum: "WEB-SOBA: Word Embeddings-Based Semi-automatic Ontology Building for Aspect-Based Sentiment Classification"
_eswc-conferences.org/ESWC/2021/Conference/Research_Track — ESWC 2021 Research_

### Official Review · AnonReviewer2 · 2021-01-06
**Simple yet effective and interesting**

**Rating:** 1
**Confidence:** 4
**Impact:** 3
**Design And Technical Quality:** 4

**Review:**

This paper propose a simple hybrid approach to aspect based sentiment analysis. It proposes to semi-automatically build an ontology that can assist with the hybrid classifier combining the ontology and a machine learning classifier. The key novel aspect of the work is the use of word embeddings for the construction of the ontology.

**Anonymity:**

Yes, I would like my review to remain anonymous.

**Reuse And Availability:**

4: High

**Strong Points:**

* Simple but clever an effective approach to ABSA.
* The paper is well written and easy to follow.

**Subreviewer:**

I submitted this review.

**Weak Points:**

* The approach could be deemed somewhat incremental, but I still consider it interesting and relevant to the conference.

---

> ### Author Rebuttal · Authors · 2021-01-28
>
> *Thank you for the nice words on our paper.*
>
> The approach could be deemed somewhat incremental, but I still consider it interesting and relevant to the conference.
>
> *The approach is indeed building on our previous experiences on semi-automatic approaches for domain sentiment ontology construction using word/synset co-occurences, but, now, using word embeddings, which involved careful consideration of several design choices and proposal of new techniques.*

---

> > ### Comment · AnonReviewer2 · 2021-02-03
> > **Thanks**
> >
> > Thanks for the comment.

---

### Official Review · AnonReviewer1 · 2021-01-11
**A semi-automatic ontology builder for aspect-based sentiment analysis**

**Rating:** 1
**Confidence:** 3
**Impact:** 3
**Design And Technical Quality:** 4

**Review:**

This paper proposed a semi-automatic ontology builder named WEB-SOBA for aspect-based sentiment analysis with respect to a specific domain using domain-specific word embeddings. The approach is evaluated with a hybrid aspect-based sentiment analysis (ABSA) framework on a benchmark dataset - the SemEval-2016 restaurant dataset. The results show that WEB-SOBA requires less time for building a domain ontology compared to other semi-automatic alternatives, and improves the performance of aspect-based sentiment classification accuracy significantly when being used in the hybrid ABSA framework.

In the following, I have some detailed comments and questions that would be taken into account to improve the paper.

Sec. 4: Instead of "User" for headings for each paragraph for user intervention in the ontology building process, it is better to have more informative headings with respect to an intervention in the corresponding step.

Sec. 4.2: It would be better to indicate that this work does not model "neutral" sentiment when the used ontology is introduced in Sec. 4.2 instead of mentioning it in Sec. 4.4

Eq: 4.5: $TH$ is a threshold used for selecting words to present to a user so that he/she can accept/reject those words. But there is already "accepted" (the number of accepted terms) in the $TH$, which makes me confused.

In the last paragraph of 4.3 says "if the cosine similarity is larger than a certain threshold, the word is added to the ontology", but in the last paragraph of page 7, it says "For each of these terms, the 15 most similar words obtained through word embeddings are suggested to the user. The user can then accept or reject adding these words as an extra lexicalization of the corresponding class. T". I'm confused that it is automatically selected via the threshold 0.7 or the user is actually accepting/rejecting based on recommendations based on word embeddings or am I missing something here.


**Questions**

- P7 - last paragraph: Is it going to make any difference by using 10 or 20 instead of 15? Is automatically adding n similar words making big difference compared to accepting/rejecting those by a user?
- P8 - Eq. 4.3: Is v_a the word embeddings of words such as "restaurant", "location"? how about "style and options"?
- P11 - Sec. 4.5: In the sentence "These clusters are ...", are those clusters refer to classes in Fig.2, if yes, where is "Quality" ? "Food Options" and "Style" also makes me confused, where in Fig. 2 there are "Food" and "Style and Options"
- P11 - Eq. 4.7: A_m and B_m are not explained
- Evaluation: Is  the user time based on a single user?

**Minor Comments**

- Related work: "Semantic Sentiment Analysis of Twitter" ISWC'12 is also an earlier work on semantic sentiment analysis although it is not ABSA
- Fig. 2.: remove (3).png
- P9: larger then -> larger than
- P10: The set of negative words is as follows: P -> N ... N- > P


**After Rebuttal**

Thanks for the authors' response for those questions. I think short descriptions regarding those points as provided in your rebuttal would make the paper more clear.

**Anonymity:**

Yes, I would like my review to remain anonymous.

**Reuse And Availability:**

4: High

**Strong Points:**

- semi-automatic domain ontology builder for ABSA, which requires less user time
- evaluation in terms of time requirements for both user and computer, and that with respect to ABSA where using the constructed ontology alone as well as using a hybrid framework with the ontology

**Subreviewer:**

I submitted this review.

**Weak Points:**

- It is not clear whether we can get the same results for other domains beyond the one evaluated in this paper
- In some places, the presentation is not clear which might make users confused

---

> ### Author Rebuttal · Authors · 2021-01-28
>
> *Thank you for the nice words on our paper.*
>
> Sec. 4: Instead of "User" for headings for each paragraph for user intervention in the ontology building process, it is better to have more informative headings with respect to an intervention in the corresponding step.
>
> *Indeed “User Intervention” is a better header than “User”, we will make this update.*
>
> Sec. 4.2: It would be better to indicate that this work does not model "neutral" sentiment when the used ontology is introduced in Sec. 4.2 instead of mentioning it in Sec. 4.4
>
> *Good point, we will move the explanations from Sec. 4.4 to Sec. 4.2 on missing “neutral” sentiment in the ontology.*
>
> Eq: 4.5:
> TH is a threshold used for selecting words to present to a user so that he/she can accept/reject those words. But there is already "accepted" (the number of accepted terms) in the TH, which makes me confused.
>
> *For each TH there is indeed a number of first presented and then accepted terms, but we select the best TH (i.e., the one that gives the best harmonic mean for the number of accepted terms and ratio of accepted terms [by the user] with respect to proposed terms [by the system]). For clarity, it would be better to use TH^star for the term on the left-hand side of “=” to indicate that this is the best threshold.*
>
> In the last paragraph of 4.3 says "if the cosine similarity is larger than a certain threshold, the word is added to the ontology", but in the last paragraph of page 7, it says "For each of these terms, the 15 most similar words obtained through word embeddings are suggested to the user. The user can then accept or reject adding these words as an extra lexicalization of the corresponding class. T". I'm confused that it is automatically selected via the threshold 0.7 or the user is actually accepting/rejecting based on recommendations based on word embeddings or am I missing something here.
>
> *These are two different steps in the methodology. On page 7 we explain how the skeletal ontology is initialized and there indeed the 15 most similar words based on word embedding cosine with respect to the pre-specified Type-1 sentiment words are suggested to the user for validation. In the last paragraph of Sec. 4.3 we explain that the user needs to check a proposed term to the ontology (which extends the skeletal ontology) and then, the other words that have a cosine similarity higher than 0.7 with the checked term are automatically added to the ontology (due to the high similarity and the fact that the user already checked the proposed term, an extra manual check on similar words is not performed).*
>
> Questions
>
> P7 - last paragraph: Is it going to make any difference by using 10 or 20 instead of 15? Is automatically adding n similar words making big difference compared to accepting/rejecting those by a user?
>
> *Based on our experimental results the 15 closest terms gave us a better trade-off between the quality of the added words and the class lexical coverage, than 10 or 20.*
>
> P8 - Eq. 4.3: Is v_a the word embeddings of words such as "restaurant", "location"? how about "style and options"?
>
> *Yes, v_a is the word embeddings of words such as “restaurant” and “location”. For Style&Options we consider the words that appear as lexicalization for Style&Options and these are words such as “style” and “options.”*
>
> P11 - Sec. 4.5: In the sentence "These clusters are ...", are those clusters refer to classes in Fig.2, if yes, where is "Quality" ? "Food Options" and "Style" also makes me confused, where in Fig. 2 there are "Food" and "Style and Options"
>
> *Indeed we need to correct the list to make it consistent with the one from page 8: Restaurant, Location, Food, Drinks, Price, Experience, Service, Ambiance, Quality and Style&Options.*
>
> P11 - Eq. 4.7: A_m and B_m are not explained
>
> *A and B are two clusters, and the m subscript is not needed. We will add the explanation.*
>
> Evaluation: Is the user time based on a single user?
>
> *Indeed one user with domain expertise was involved in the validation step.*
>
> Minor Comments
>
> Related work: "Semantic Sentiment Analysis of Twitter" ISWC'12 is also an earlier work on semantic sentiment analysis although it is not ABSA
>
> *We have focused on aspect-based sentiment classification as the main topic of the paper due to the limited number of pages.*
>
> Fig. 2.: remove (3).png
>
> *We will perform the correction.*
>
> P9: larger then -> larger than
>
> *We will perform the correction.*
>
> P10: The set of negative words is as follows: P -> N ... N- > P
>
> *We will perform the correction.*

---

### Official Review · AnonReviewer4 · 2021-01-11
**Sensible but unsurprising iterative improvement on previous work**

**Rating:** 1
**Confidence:** 4
**Impact:** 2
**Design And Technical Quality:** 3

**Review:**

The paper tackles the problem of semi-automatically building ontologies for aspect-based sentiment classification, i.e., classifying the sentiment of a given sentence with regard to a given aspect.
Previous work tackled the same problem using word co-occurrences, which the paper under review proposes to substitute for word embeddings.
The idea being that word embeddings would better capture word semantics than comparing word co-occurrences would, which should thereby result in a better model, similar to how word embeddings have been applied to so many other tasks.
The resulting ontologies outperforms semi-automatically created ontologies from previous approaches by a decent margin.
Overall, the proposed idea is a sensible iterative improvement upon previous work.
The papers presentation is good, everything should be understandable even to outsiders of the respective field.
Data and the implementation is made public in a GitHub repository.
I am therefore confident that the results of the paper are easily reproducible.

On the downside, the presented model seem so straightforward and comparable to other existing approaches that this work overs basically no take-away insights into the topics of word embeddings, ontology construction, or sentiment analysis, besides that it now seemingly constitutes the best approach for this very specific task.
For example, while the authors measured that creating an ontology after this methodology requires less user interaction time than previous approaches, they offer no insight into why exactly this is the case, i.e., what this approach does better than the previous word co-occurrence based one.
[Popular work that empirically questions the benefits of word embeddings over count based methods](https://transacl.org/ojs/index.php/tacl/article/view/570) is not considered, it therefore remains unclear whether the previous co-occurrence-based methods could have not also just been adjusted with other, similar approaches.
Last, one could question whether a semi-automatic ontology building approach as presented in the paper is actually ultimately necessary.
According to the evaluation, building a comparable ontology fully manually takes only 7 hours for a single ontology engineer.
Considering that probably not so many aspect-based sentiment analysis domains with sufficient data to implement this approach exist, I am not sure that anyone might want to create so many ontologies that they couldn't have done that by the time they have implemented the proposed model.

As an additional baseline it would have been interesting to see what performance a traditional machine learning method, trained on the word embeddings refined for sentiment analysis, would be able to achieve on the evaluation task, to see if the ontology build for the task actually helps, or whether most of the results could also be achieved by fully-automatic means.

Minor points:

- I could not follow the example presented in section 3.
  What is a `PricesPropertyMention` or a `LocationPositiveProperty`?
  A figure with an example ontology would have been appreciated.
- To me, it is not clear what the $v_a$ refers to in eq. (4.3).
  Is it the word embedding of one of the literal representations of the classes of set $\mathcal{A}$?
- I don't understand the threshold calculation of eq. (4.5):
    - The term inside of the parentheses seems to be independent over the different values the maximum iterates over.
    - I don't understand how something can be defined in a pattern of $x = \max_x(y)$, i.e., variable being defined is bound inside of the $\max$.
    - I don't understand how values of *number of suggested* and *accepted terms* can be used to find the threshold, as from my understanding the threshold would be used to find out what to suggest to the user.
    - I don't know what is meant by "lexical class".

Typos:
- Fig 2.: "(3).png"
- Section 4.4: Definition of $\mathcal{P}$ and $\mathcal{N}$ seem to be the wrong way around.

**Anonymity:**

Yes, I would like my review to remain anonymous.

**Reuse And Availability:**

4: High

**Strong Points:**

- Sensible straightforward iteration on previous work.
- Method that outperforms previous approaches by a decent margin.
- Good presentation.
- Implementation and data publicly available.

**Subreviewer:**

I submitted this review.

**Weak Points:**

- Paper offers no insights into why it performs better than previous approaches.
- Paper offers little take-away insights for the respective fields of study.
- Unclear whether such a semi-automatically approach is actually needed in practice.

---

> ### Author Rebuttal · Authors · 2021-01-28
>
> *Thank you for the nice words on our paper.*
>
> On the downside, the presented model seem so straightforward...
>
> *Converting our previous approach SASOBUS based on synset co-occurences to our current approach WEB-SOBA based on word synsets was far from trivial. Properly defining the DomainSimilarity and MentionClassSimilarity as replacements for Domain Pertinence and Domain Consensus required novel solutions. Same goes for the Positive Score (PS) and Negative Scored (NS) attached to a sentiment carrying mention, defined as well in a word embedding context. In addition producing an aspect hierarchy needed us to adapt agglomerative clustering to make sure that hierarchies are built for each aspect type. These are just a few examples of new solutions that we bring to the literature on how to employ word embbeddings for aspect-based sentiment classification (independent of the considered domain).*
>
> For example, while the authors measured that creating an ontology...
>
> *In Table 1 we have shown that the amount of lexicalizations is smaller for WEB-SOBA than the other approaches which is the result that fewer terms but of a better quality (as shown later in the paper) were generated by this approach. The obtained terms seem to better represent the considered domain.*
>
> Popular work that empirically questions...
>
> *Indeed some papers debate the performance advantages of word embeddings over word co-occurences. Here we exploit a clear benefit of word embeddings, i.e., the reduced dimensionality of the word embeddings compared to the co-occurrence vectors speeding calculations and requiring a smaller memory footprint. The fact that we also obtain better performance with word embeddings was a nice result which we did not know a priori.*
>
> Last, one could question whether a semi-automatic ontology building approach...
>
> *Our solution relies on the existence of text for a certain domain which is not very demanding. There are many datasets or easy to collect ones that are specific to a certain domain. Please note that these are given as plain text, we do not require any annotations. The 7 hour of user work for the manual approach should be compared with the 40 minutes of user work required WEB-SOBA. Even if we have only 1 domain it is better to use WEB-SOBA to build the ontology than building it manually from scratch. Given that our approach is domain-independent and to some extent also language independent the benefits of using WEB-SOBA for aspect-based sentiment classification are clear.*
>
> As an additional baseline...
>
> *We have already done this experiment in our previous work quoted in the current paper [27] where we have shown that the neural network approach using word embeddings is inferior to the hybrid approach that uses ontology reasoning in the first step and the very same neural network as backup in the second step.*
>
> Minor points:
>
> I could not follow the example presented in section 3...
>
> *PricesPropertyMention is a class that represents a Property (adjective) with respect to Prices (e.g., expensive, cheap). LocationPositiveProperty is a class that represents a Property as well, but for which we additionally know that it is denoting Positive sentiment. LocationPositiveProperty is a subclass of LocationPropertyMention and Positive classes.*
>
> To me, it is not clear what the v_a refers to in eq. (4.3)...
>
> *v_a refers to the vector associated to the term a from the aspect mention class A (e.g., a can be ‘’restaurant” from the aspect mention class Restaurant).*
>
> I don't understand the threshold calculation of eq. (4.5):
> ...
> I don't know what is meant by "lexical class".
>
> *The lexical class is one of the considered part-of-speeches (noun, verb, or adjective). The argument of the max function is the threshold TH_lc which is the threshold (TH) associated to the current lexical class (lc) [we have three best thresholds, one per lexical class]. The term in parenthesis depends on the current threshold as for each threshold value different terms are presented to the user for validation and n, the number of suggested terms, and, accepted, the number of accepted terms, depend on the current threshold (e.g., higher threshold implies less terms to be presented to the user and thus less checks needed by the user). The output of the argument of the max function (the best threshold) is used to set the threshold value, i.e., we select the threshold which has the highest value for the term in parentheses (i.e., best harmonic mean for the number of accepted terms and ratio of accepted terms [by the user] with respect to proposed terms [by the system]). For clarity, it would be better to use TH^star for the term on the left-hand side of “=” to indicate that this is the best threshold.*
>
> Typos:
>
> Fig 2.: "(3).png"
>
> *We will perform the correction.*
>
> Section 4.4: Definition of P and N seem to be the wrong way around.
>
> *We will perform the correction.*

---

> > ### Comment · AnonReviewer4 · 2021-01-28
> > **Response to Rebuttal**
> >
> > > > On the downside, the presented model seem so straightforward...
> > >
> > > Converting our previous approach SASOBUS based on synset co-occurences to our current approach WEB-SOBA based on word synsets was far from trivial. Properly defining the DomainSimilarity and MentionClassSimilarity as replacements for Domain Pertinence and Domain Consensus required novel solutions....
> >
> > Both $DS_i$ and $MCS_i$ are defined as cosine similarities, which are IMO very natural choices in the word embedding context. Of course I will concede that I did not implement this system, so there is probably  be a lot that I'm missing. However, it still stands that the paper offered little take-away insights for me.
> >
> > > > For example, while the authors measured that creating an ontology...
> > >
> > > In Table 1 we have shown that the amount of lexicalizations is smaller for WEB-SOBA than the other approaches which is the result that fewer terms but of a better quality (as shown later in the paper) were generated by this approach. The obtained terms seem to better represent the considered domain.
> >
> > I don't understand how this answers my point. It might explain *what* the output of your approach is better at, but still not *why*.
> >
> > > > Popular work that empirically questions...
> > >
> > > Indeed some papers debate the performance advantages of word embeddings over word co-occurences. Here we exploit a clear benefit of word embeddings, i.e., the reduced dimensionality of the word embeddings compared to the co-occurrence vectors speeding calculations and requiring a smaller memory footprint. The fact that we also obtain better performance with word embeddings was a nice result which we did not know a priori.
> >
> > I fear this is missing the point I was trying to make. Of course, word embeddings have advantages over word co-occurrence matrices. However, the linked paper shows that *factorizing* word co-occurrence matrices reaches performance that is similar or better than word2vec. So I was wondering where the benefit to previous co-occurrence based ontology building approaches would be, *if* one would employs such a factorization (if that is possible).
> >
> > > > Last, one could question whether a semi-automatic ontology building approach...
> > >
> > > Our solution relies on the existence of text for a certain domain which is not very demanding. There are many datasets or easy to collect ones that are specific to a certain domain. Please note that these are given as plain text, we do not require any annotations. The 7 hour of user work for the manual approach should be compared with the 40 minutes of user work required WEB-SOBA. Even if we have only 1 domain it is better to use WEB-SOBA to build the ontology than building it manually from scratch. Given that our approach is domain-independent and to some extent also language independent the benefits of using WEB-SOBA for aspect-based sentiment classification are clear.
> >
> > Yes, *once* the presented semi-automatic system is implement or adapted to the domain of interest, one only needs to spent 40 minutes to build the ontology. My guess is that this set-up time is probably far larger than just 7 hours. Still, I feel that the paper has a point, which is why I did not rate as reject.
> >
> > > > As an additional baseline...
> > >
> > > We have already done this experiment in our previous work quoted in the current paper [27] where we have shown that the neural network approach using word embeddings is inferior to the hybrid approach that uses ontology reasoning in the first step and the very same neural network as backup in the second step.
> >
> > From a quick glance through the references of [27] it seems like only GLoVe was used in that paper, and not a word embedding refined to be sentiment-aware, which would be the correct comparison. Or am I missing something here?
> >
> > > > I could not follow the example presented in section 3...
> > >
> > > PricesPropertyMention is a class that represents ...
> > >
> > > > To me, it is not clear what the v_a refers to in eq. (4.3)...
> > >
> > > v_a refers to the vector associated to ...
> > >
> > > > I don't understand the threshold calculation of eq. (4.5): ... I don't know what is meant by "lexical class".
> > >
> > > The lexical class is one ...
> >
> > Thanks for those in-depth explanations! However, when I say "I don't understand" in a review (in the minor points), I don't necessarily mean that I care about what the correct explanation is. Instead, I mostly want to indicate the the wording *in the paper* should probably be updated because it is ambiguous or confusing.
> > For example, I could piece-together what $v_a$ probably means, but it is not defined in the paper!

---

### Official Review · AnonReviewer5 · 2021-01-14
**Semi-automatic sentiment classification for restaurant reviews**

**Confidence:** 4
**Impact:** 2
**Design And Technical Quality:** 3

**Review:**

I have read the authors' rebuttal which clarifies some points but does not alter my review.
-------
The paper describes an approach to semi-automatically build an ontology for aspect-based sentiment classification of restaurant reviews. The paper is well-written and the presented experiment seems to be performed well, however, the paper lacks an explanation for certain methodological design choices, a qualitative error analysis and it not yet clear how well the approach generalises to other domains. Furthermore, the paper is building on previous work by a subset of the authors, but this is not made clear. For example Dera, E., Frasincar, F., Schouten, K., Zhuang, L.: Sasobus: Semi-automatic sentiment domain ontology building using synsets. In: European Semantic Web Conference. pp. 105–120. Springer (2020) uses a similar setup as WEB-SOBA but focuses on the use of synsets instead of word embeddings, of course it is only normal that authors build on their prior work, but considering the results differ only minimally, with the previous approach even outperforming the approach presented in this paper on the SemEval-16 dataset, it would be good if the authors could explain more explicitly the added value of the new approach.

Text comments:
Abstract
- domain specific -> domain-specific

Introduction:
- It is of course perfectly OK to use a dataset where aspects are already indicated, but it would also be interesting to see the impact of an end-to-end system and automatically identified (and thus potentially not perfect) aspects. Perhaps the authors could reflect on that in the conclusion wrt future work?
- describe our proposal methodology -> describe our methodology

Section 2:
- delete 'well-known' (2x)
- just because word2vec is the most widely used doesn't mean it is the best fit for your research problem. Since the conclusion also mentions that BERT and ELMo could be used, I wonder why different methods weren't tried. In any case, I think it would be good if the authors better motivate their choice for word2vec.

Section 3:
- leaving us with -> resulting in
- what is meant by 'the contrasting corpus'?
- Each sentence contains opinions ... negative, neutral, positive} -> this explanation could use a bit more space, now it can be difficult to follow for readers less familiar with sentiment analysis
- Fig 1 is a bit hard to read

Section 4:
- This model is trained using the following loss function -> please motivate the choice for this loss function
- Fig 2: remove (3).png
- Adverbs can also be considered for sentiment information, but they affect the intensity of the sentiment rather than the polarity -> how about 'atrociously presented', or 'rudely served'?
- After selecting all the important terms -> delete 'all' (how do you know you have all of them, was this step evaluated separately?)

Section 5:
- Please explain the Welch t-test and how to interpret the provided values in Tables 3 and 4
- A qualitative evaluation is missing, therefore it is now unclear for what types of sentiments or aspects the approach works best and for which further research is needed.


**Anonymity:**

No, I would like my review to be deanonymized.

**Rating:**

-2: Reject

**Reuse And Availability:**

3: Medium

**Strong Points:**

- well-written
- relevant problem
- experiments seems to have been carried out properly

**Subreviewer:**

I submitted this review.

**Weak Points:**

- only minor improvement over prior work
- generalisability not researched
- quantitative error analysis lacking
- motivation for experimental design choices lacking

---

> ### Author Rebuttal · Authors · 2021-01-28
>
> *Thank you for the nice words on our paper.*
>
> For example Dera, E., Frasincar, F., Schouten, K., Zhuang, L.: Sasobus: Semi-automatic sentiment domain ontology building using synsets ... but considering the results differ only minimally, with the previous approach even outperforming the approach presented in this paper on the SemEval-16 dataset, it would be good if the authors could explain more explicitly the added value of the new approach.
>
> *Indeed the setup is similar as in our previous work on SASOBUS, but with the required adaptation for using word embeddings instead of synset co-occurences. In both ontology only (Table 3) and hybrid model (Table 4) out-of-sample our current approach WEB-SOBA is better than SASOBUS. In addition, for the hybrid model, using in-sample cross-validation WEB-SOBA is statistically significant better than SASOBUS. For the ontology only, using in-sample cross-validation the two methods are not statistically different.*
>
> Text comments: Abstract
>
> domain specific -> domain-specific
>
> *We will perform the correction.*
>
> Introduction:
>
> It is of course perfectly OK to use a dataset where aspects are already indicated, but it would also be interesting to see the impact of an end-to-end system and automatically identified (and thus potentially not perfect) aspects. Perhaps the authors could reflect on that in the conclusion wrt future work?
>
> *Our focus here is on aspect-based sentiment classification. Systems that aim to tackle both problems: aspect detection and sentiment classification at once usually score lower for sentiment classification than systems that assume the gold aspects to be given, which is as expected. We have discussed this in our survey: “Kim Schouten, Flavius Frasincar: Survey on Aspect-Level Sentiment Analysis. IEEE Trans. Knowl. Data Eng. 28(3): 813-830 (2016)”, which we refer to also in this paper [22].*
>
> describe our proposal methodology -> describe our methodology
>
> *We will perform the correction.*
>
> Section 2:
>
> delete 'well-known' (2x)
>
> *We will perform the correction.*
>
> just because word2vec is the most widely used doesn't mean it is the best fit for your research problem. Since the conclusion also mentions that BERT and ELMo could be used, I wonder why different methods weren't tried. In any case, I think it would be good if the authors better motivate their choice for word2vec.
>
> *Next to the fact that word2vec is the most popular tool for word embedding we have indicated in the paper that it is “training time efficient” as differently than GloVe it can be trained in an online fashion while GloVe needs to update the co-occurrence matrix when new instances are presented. We agree with the reviewer that using a popular tool is not a strong argument and we plan to replace it with the second advantage of having a small memory footprint.*
>
> Section 3:
>
> leaving us with -> resulting in
>
> *We will perform the correction.*
>
> what is meant by 'the contrasting corpus'?
>
> *In Section 3 we have explained that the contrasting corpus is google-news-300. This is a contrastive corpus as it does not focus on our considered domain, the restaurant domain.*
>
> Section 4:
>
> This model is trained using the following loss function -> please motivate the choice for this loss function
>
> *The loss function is the one used by word2vec CBOW and explained in references [15] and [16].*
>
> Fig 2: remove (3).png
>
> *We will perform the correction.*
>
> Adverbs can also be considered for sentiment information, but they affect the intensity of the sentiment rather than the polarity -> how about 'atrociously presented', or 'rudely served'?
>
> *Here we refer to intensifying adverbs like “absolutely” in “absolutely no” and “absolutely delighted” that focus on intensity and not sentiment. We did mention [10] the work that presented this study. The reviewer is right, some adverbs can carry sentiment like in the examples given, we need to replace “adverbs” by “intensifying adverbs”.*
>
> After selecting all the important terms -> delete 'all' (how do you know you have all of them, was this step evaluated separately?)
>
> *’all’ refers to adjectives, nouns, and, verbs as given by the used part-of-speech tagger.*
>
> Section 5:
>
> Please explain the Welch t-test and how to interpret the provided values in Tables 3 and 4
>
> *As explained in the paper, the Welch t-test is used to compare the means of the cross-validation results for two methods. It is more reliable than the Student t-test as the samples do not need to be paired, i.e., the folds can be computed differently for each method. If the reported p-value is below 0.05 we consider the means to be statistically different and obeying the ordering given by their sizes.*
>
> A qualitative evaluation is missing, therefore it is now unclear for what types of sentiments or aspects the approach works best and for which further research is needed.
>
> *This is an excellent point, we did not do it and can be included in future work.*

---

### Official Review · AnonReviewer3 · 2021-01-17
**WEB-SOBA Review - Convincing results in term of computation time but more detail is needed to reproduce them**

**Rating:** 2
**Confidence:** 4
**Impact:** 3
**Design And Technical Quality:** 4

**Review:**

This paper describes an approach for semi-automatically building a sentiment/aspect ontology using word embeddings. The ontology is evaluated for the task of aspect-based sentiment analysis with a use case in the restaurant domain.

The paper is well written, easy to follow and reasonably structured.

Results show reduced computation time and less user involvement compared to other approaches based on word co-occurence, while achieving similar or better performance.

In general the approach is described with enough detail to reproduce the results but more information is needed for constructing the "domain-related word embedding model" for term extraction. It is not clear how this was computed.

Missing reference for domain modeling:
Bordea, Georgeta, Paul Buitelaar, and Tamara Polajnar. "Domain-independent term extraction through domain modelling." In The 10th international conference on terminology and artificial intelligence (TIA 2013), Paris, France. 10th International Conference on Terminology and Artificial Intelligence, 2013.

**Anonymity:**

Yes, I would like my review to remain anonymous.

**Reuse And Availability:**

4: High

**Strong Points:**

- Reduced computation time
- Smaller need for user involvement
- Comparable results with the state of the art

**Subreviewer:**

I submitted this review.

**Weak Points:**

- Clarification needed about how to use word-embeddings to build a domain model
- Missing reference

---

> ### Author Rebuttal · Authors · 2021-01-28
>
> *Thank you for the nice words on our paper.*
>
> In general the approach is described with enough detail to reproduce the results but more information is needed for constructing the "domain-related word embedding model" for term extraction. It is not clear how this was computed.
>
> *In Section 3 we have explained that we use Yelp Open Dataset and google-news-300 with word2vec CBOW to obtain word embeddings for the domain corpus and the contrasting corpus, respectively.*
>
> Missing reference for domain modeling: Bordea, Georgeta, Paul Buitelaar, and Tamara Polajnar. "Domain-independent term extraction through domain modelling." In The 10th international conference on terminology and artificial intelligence (TIA 2013), Paris, France. 10th International Conference on Terminology and Artificial Intelligence, 2013.
>
> *This is interesting work that we were not aware of. We plan to include it in the future work of the paper as domain modelling complements well our contrastive corpus-based solution by providing domain specific terms that are more generic than the current ones.*

---

### Decision · Program_Chairs · 2021-02-23

**Decision:**

Accept with shepherding

**Comment:**

This meta review summarizes the strengths and weaknesses pointed out by the reviewers. All reviewers agreed that the paper addresses an interesting and important problem. There was also consensus that the paper is well written. One weakness identified by several reviewers is the fact that the paper appears to be highly incremental when compared to some of the authors' own prior work. Also, the choice of word2vec CBOW instead of other more recent word-embedding approaches was questioned and seen as requiring better justification. Lastly, the experimental evaluation was criticized for its focus on a single domain, leaving the generalizability an open issue, and for its lack of a qualitative evaluation. For the paper to be accepted, the authors are kindly asked to address the following issues and submit a revised version:

[Task 1] Include a more detailed description of what constitutes the technical deltas and improvements over the earlier work [8] (cf. Review 5).

[Task 2] Justify the choice of word2vec CBOW over more recent word-embedding approaches (e.g., GloVE, FastText, BERT, ELMo) and the choice of your loss function (cf. Review 5).

[Task 3] Extend the experimental evaluation by a qualitative evaluation. This could be driven by anecdotal examples, for which your approach wins/looses in comparison to the competitors due to its use of word embeddings. (cf. Review 5)

[Task 4] Discuss the generalizability of the proposed approach. What would be required for a different domain (e.g., movies or hotels) to apply the approach? This should also address the question (cf. Review 4) whether it would be worthwhile at all to employ the proposed approach instead of a manual curation, when factoring in time for preparing the data and deploying the code.

[Task 5] Address reviewers' comments regarding small presentation issues and missing technical details (e.g., Welch t-test).

Please submit the revised version of the paper by March 17th 2021. Should you have any questions or want feedback on your changes, feel free to contact Klaus Berberich (klaus.berberich@htwsaar.de).